# Domain Adaptation with Augmented Data by Deep Neural Network Based Method Using Re-Recorded Speech for Automatic Speech Recognition in Real Environment

**DOI:** 10.3390/s22249945

**Published:** 2022-12-16

**Authors:** Raufun Nahar, Shogo Miwa, Atsuhiko Kai

**Affiliations:** 1Graduate School of Science and Technology, Shizuoka University, Hamamatsu 432-8561, Japan; 2Graduate School of Integrated Science and Technology, Shizuoka University, Hamamatsu 432-8561, Japan

**Keywords:** ASR, real environment, data augmentation, DNN, feature transformation, recording alignment, VoLTE, classroom recording

## Abstract

The most effective automatic speech recognition (ASR) approaches are based on artificial neural networks (ANN). ANNs need to be trained with an adequate amount of matched conditioned data. Therefore, performing training adaptation of an ASR model using augmented data of matched condition as the real environment gives better results for real data. Real-world speech recordings can vary in different acoustic aspects depending on the recording channels and environment such as the Long Term Evolution (LTE) channel of mobile telephones, where data are transmitted with voice over LTE (VoLTE) technology, wireless pin mics in a classroom condition, etc. Acquiring data with such variation is costly. Therefore, we propose training ASR models with simulated augmented data and fine-tune them for domain adaptation using deep neural network (DNN)-based simulated data along with re-recorded data. DNN-based feature transformation creates realistic speech features from recordings of clean conditions. In this research, a comparative investigation is performed for different recording channel adaptation methods for real-world speech recognition. The proposed method yields 27.0% character error rate reduction (CERR) for the DNN–hidden Markov model (DNN-HMM) hybrid ASR approach and 36.4% CERR for the end-to-end ASR approach for the target domain of the LTE channel of telephone speech.

## 1. Introduction

The study of automatic speech recognition (ASR) is a vast area of artificial intelligence (AI). ASR is important for AI as much as hearing capability for humans. ASR enables the machines and software to understand the representations of sound computationally. The representations can be fragments of a second of speech data, also known as a frame, a word, or a sequence of words. Depending on the purpose, the ASR can be equipped to perform tasks such as converting speech to text, a well-known application which enables us to keep record of meetings and conversations, automatic caption generation for people with hearing impairment and so on.

In one of the long-standing ASR approaches, ASR models consist of two vital components. One is the language model, which deals with the linguistic aspects of speech, and the other is the acoustic model to process acoustic aspects of speech. Conventionally, acoustic models are trained using large-scale clean data recorded in ideal condition—a noise-free environment with a high-quality close-talking microphone and so on. These acoustic models perform well for recognizing clean speech data. However, they do not perform as effectively for data that are recorded in conditions with variability. The quality of data affects the performance of ASR system. The recording condition of data can range from simple environments such as a classroom equipped with a regular wireless pin mic to complex environment such as telephone channel recording, which can involve countless types of handsets, transmission networks and background noise at the location of the caller and receiver. Therefore, it is necessary to improve the performance of acoustic models for real environment speech data in terms of cost-effectiveness and convenience.

There is previous research on how to improve the acoustic model’s performance for real data. According to it, one method to achieve improvement is to perform domain adaptation. To perform domain adaptation for real-world data, data augmentation is often used. Previous research on this subject shows data augmentation by adding various recorded noises at the desired SNR level, performing speed perturbation on clean data, or using room impulse responses (RIR) to simulate the desired room acoustics. These augmented data contribute to producing improved performance of ASR systems for the target domain, as stated in [1,2,3,4]. Moreover, there are DNN-based data augmentation methods which extract acoustic information to represent acoustic environments instead of creating simulated data, which are explained in [5].

The general theory is that acoustic models need to be trained with relevant data to achieve the best performance. Real-world recording media are bound to vary in terms of recording conditions, transmission channels, etc. Therefore, it is important to prepare enough data for each condition with proper transcription for training an acoustic model appropriate for the environments accordingly, as described in [6,7,8,9]. Acquiring and preparing such data can be expensive and time consuming.

In cases of matched domain data scarcity, domain adaptation is a well-accepted solution. In [10], the data scarcity problem is addressed. The speech recognition performance for low-resource data of dysarthric speech is improved by training a statistical-based Gaussian mixture model (GMM) using high-resource meeting speech data. Then, a maximum a posteriori (MAP) training adaptation approach is used for fine-tuning the ASR using small-scale dysarthric speech data. Another research addressing label scarcity in target domain is described in [11]. This research states the effectiveness of DNN-based data augmentation, where more data are generated for the target domain from the transcribed clean data using a variational autoencoder (VAE). The present research takes inspiration from the research stated above to perform domain adaptation by involving a fine-tuning approach using data augmented by a simple DNN-based model, which is much more cost-effective than a rather complex LSTM-based VAE. We propose a novel method of data augmentation that helps to imitate the real environment by using a feed-forward regression model. Similar approaches have been taken to perform denoising or dereverberation as preprocessing in [12], in reverse. To the best of our knowledge, most of the research are performed to increase the size of the training data for ASR as much as possible. However, using larger data also increases computational cost. Therefore, in this research, small amounts of data are taken into consideration to train the regressing model to learn on pairs of clean re-recorded data of very small size (the smallest with a duration of 0.2 h). Experimental results show that it successfully transforms clean input data to real-environment output data by feature transformation. Since it learns to transform features directly from real-environment data, the computation cost is also minimal. The augmented data that we prepare in this way are then used to perform training adaptation of acoustic models by the fine-tuning method. To acquire the best result, the original re-recorded speech is also used along with the transformed features.

Moreover, data acquisition by re-recording is also an important part of this research. In the case of supervised training of acoustic models, proper segmentation and transcriptions are needed. Acquiring the desired quality data in real environments is difficult. Even if spontaneous data are acquired, preparing segments and texts accordingly is more difficult. To overcome this difficulty, re-recording of speech from an official data corpus with proper transcription is proposed. In this way, data of the desired real environment, such as telephone channels, classroom data, etc., can be created. In this research, spontaneous monologue speech recordings, played through a loudspeaker at the caller’s end, are recorded through telephone channels. Another variation is classroom data. Monologue speeches are also re-recorded in similar fashion with necessary adjustments for classroom environment with a normal low-quality wireless pin mic. The datasets are used in the experiments with the proposed models.

The main contributions of the present work are as follows:A domain adaptation approach independent of the ASR model is proposed by preparing speech data which contain target domain characteristics. Experiments are conducted on two of the state-of-the-art approaches, and the effectiveness of the proposed method on both of them is proved.To prepare speech data with target domain characteristics at low cost, the following approaches are adopted:Use an already existing corpus.Re-recording the corpus data by playing them in a real environment for only a limited short period of time.Performing post-processing on them to adapt them for ASR model training.It is proved that by involving a simple regression model for transforming, it is possible to obtain data with target domain characteristics from clean data if only a small amount of target domain data (duration of less than 1 h) is acquired to improve the performance of ASR to a great extent.

This paper explains research about fine-tuning-based domain adaptation for a robust automatic speech recognition system. In Section 2, the ASR models used for domain adaptation in this research, data acquisition techniques of real environments, the problems we face with the data and the proposed method of solving those problems are explained. In the latter half of Section 2, the task setting for this research along with the proposed DNN-based data augmentation method and overall data augmentation approaches, are explained. In Section 3, the experimental setups including preparation and usage of datasets and neural network (NN) models used in this research are described. Section 4 focuses on explaining results and discussion. Finally, the research is concluded in Section 5.

## 2. Materials and Methods

### 2.1. ASR Models for Domain Adaptation

The task of automatic speech recognition (ASR) infers word sequences for given input speech. Traditionally, an ASR system consists of various modules, such as acoustic model, language model, decoder, etc. In this research, we investigate domain adaptation in two different ASR frameworks: a DNN-HMM hybrid ASR framework and an end-to-end ASR framework. A DNN-HMM hybrid ASR system consists of an acoustic model $P_{AM}\left(X|W\right)$ and an explicit language model $P_{L}\left(W\right)$, which are used for searching the most likely $\hat{W}$ which maximizes the probability of speech *X* being a phoneme, character or word sequence P(W|X) with the help of the language model. The DNN-HMM acoustic model is described in Section 2.1.1. The other approach of ASR explored in this research is a hybrid connectionist temporal classification (CTC)/attention end-to-end (E2E) model, which usually directly predicts character sequence probability $P_{E2E}\left(W|X\right)$ without the help of an explicit language model, unlike the DNN-HMM hybrid model [13,14,15,16,17]. The end-to-end model consisting of encoder–decoder is described in Section 2.1.2.

#### 2.1.1. DNN-HMM-Based Speech Recognition

As acoustic model, context-dependent phone-unit (triphone) HMMs are used to estimate $P_{AM}\left(W|X\right)$, where *W* is phoneme, character or word sequence when speech feature *X* is given. The DNN-HMM hybrid ASR framework comprises the DNN-HMM acoustic model to predict $P(s|x_{t})$, where *s* is the posterior probability of HMM state when speech feature $x_{t}$ is given at each time frame *t*, as presented in Equations (Equation 1) and (Equation 2) [7]. To train the DNNs, the state labels of the HMM corresponding to each time of the input signal are used as the teacher signal. This allows us to learn a nonlinear function that estimates the posterior probability $P(s|x_{t})$ of each phone-state *s* with respect to the input features. The output of the DNN can then be transformed into the likelihood value of the phone-state class by Bayes’ theorem.
(1)$$P\left(x_{t}|s\right)=\frac{P\left(s|x_{t}\right)P\left(x_{t}\right)}{P(s)},$$
(2)$$\;\propto \frac{P(s|x_{t})}{P(s)},$$
Here, $P(x_{t})$ is omitted since it does not affect the optimization. P(s) is obtained from the frequency of occurrence of the correct target label for the training data. Speech recognition using DNN-HMM is performed by replacing the calculation of the output distribution of HMM by transformation in the form of $P(x_{t}|s)$.

As the DNN acoustic model of the DNN-HMM, we used a time-delay neural network (TDNN) [18]. A time-delay neural network is a feed-forward neural network that effectively models long-range temporal dependencies [19] in data. It exploits a modular and incremental design to create larger networks from subcomponents [20]. Traditional architectures compute the hidden activations at all time steps, which makes it computationally expensive. The architecture used in this research uses a subsampling technique, which allows the hidden activations to be computed at only a few time steps at each level [18], making it computation-friendly. Since in a DNN-HMM ASR framework, the DNN classifier is trained with the target label given as HMM state sequence $\hat{s_{t}}$, which is estimated by using the GMM-HMM acoustic model trained by using a large-scale clean dataset and its transcription, it is important for the training data to be aligned with the target as much as possible. Moreover, to obtain the best results, we extract forced alignments using clean data for the corresponding re-recorded data as the input. Therefore, the training may suffer from the internal temporal misalignment described in later sections.

#### 2.1.2. End-to-End (E2E)-Based Speech Recognition

Nowadays, due to the advancement in the field of computation, even computationally expensive methods are also being adopted to perform at low cost. One of these methods is the end-to-end speech recognition method, which encompasses multiple recognition modules into a single but rather complex and deep neural network. This framework allows the entire speech recognition model to be optimized at once. It takes acoustic features as input and outputs phoneme sequences or character or word sequences, and it does not need any explicit pronunciation dictionary or acoustic model. Recurrent neural network (RNN)-based models are mostly used for end-to-end speech recognition. There are two most popular approaches of training the RNN-based end-to-end model at present. One is connectionist temporal classification (CTC) loss-based training of the encoder–decoder model, and the other is the training of the encoder–decoder model using an attention mechanism. The method of training the encoder–decoder using the CTC objective $P_{\mathrm{CTC}}\left(C|X\right)$ and an attention objective $P_{\mathrm{Att}}^{*}\left(C|X\right)$ together is called a hybrid approach. Both of the objectives maximizes the probability of target character sequence *C* for speech feature sequence *X*. However, $P_{\mathrm{Att}}^{*}\left(C|X\right)$ takes previous target label sequence into account. A hybrid approach is expressed using the following formulae [16]:(3)$$L_{\mathrm{CTC}}=-logP_{\mathrm{CTC}}\left(C|X\right),$$
(4)H=Encoder(X),
(5)$$P_{\mathrm{Att}}^{*}\left(C|X\right)=\mathrm{Decoder}\left(H\right),$$
(6)$$L_{\mathrm{Att}}=-logP_{\mathrm{Att}}^{*}\left(C|X\right),$$
(7)$$L=\alpha L_{\mathrm{CTC}}+\left(1-\alpha \right)L_{\mathrm{Att}},$$
where *X* is the input acoustic feature sequence, and *H* is the intermediate feature sequence obtained by transforming the acoustic feature into a higher-order feature representation. $L_{\mathrm{CTC}}$ is the loss of CTC which is the negative logarithmic value of the objective function $logP_{\mathrm{CTC}}\left(C|X\right)$ and $L_{\mathrm{Att}}$ is the loss of the decoder using source–target attention, which is the negative logarithmic value of the objective function $logP_{\mathrm{Att}}^{*}\left(C|X\right)$. *L* is the loss of the whole model, which is the weighted sum of the loss of CTC and the loss of the decoder with the constant α.

By using the CTC loss function, it is possible to produce an output sequence directly from the input sequence without using the HMM. In the encoder–decoder mechanism, the encoder takes input speech feature frames and generates a fixed-length vector in an LSTM (long short-term memory) RNN at a certain frame interval. In the decoder, the output is generated via LSTM from the vector generated by the encoder. In any given time frame, the hidden state in the decoder is computed using the previous hidden layer state of the previous RNN decoder. The shortcoming of the encoder–decoder mechanism is that it can not perform well for long time series data, as it is impossible to fit all the information to a fixed-length vector. Therefore, the attention mechanism is introduced, which helps determine the hidden layer of the decoder by referring directly to the state of the encoder at each time frame, and the state most relevant to the current time frame is selected.

Since both the CTC and the attention mechanism have their advantages and shortcomings, a hybrid approach has been developed involving both CTC and attention [16]. The final output of the hybrid CTC/attention model is the sum of the output vectors of CTC and attention. The loss function during training is defined as the weighted linear sum of the loss functions of CTC and attention.

Using CTC provides the privilege of fast convergence, and the attention mechanism allows it to have the similar effect of context-dependent training by using past history. Therefore, hybrid CTC/attention is a very popular way of implementing the speech recognition system. An E2E ASR model does not require explicit HMM (phone) state alignment as does DNN-HMM ASR. Therefore, it is less likely to be affected by the external alignment extraction problem. However, the target character sequence of the training data is prepared by using the transcription information provided by the corpus. Therefore, though not in phoneme level as DNN-HMM, in the case of our experiments, the E2E model may suffer from the temporal misalignment issue to some extent in the fine-tuning phase when we use re-recorded speech as a real-world example if the segments have temporal misalignment at starting and ending. Therefore, correcting any temporal deviation at the starting and ending of the re-recorded speech can improve the performance of the end-to-end model significantly, unlike other approaches.

### 2.2. Re-Recorded Data Acquisition

#### 2.2.1. Re-Recording of Clean Data in Real Environment

Training an acoustic model to adapt to a particular environment requires acquiring data from the environment as much as possible. This problem may seem to be solved by recording data at convenience. However, this simple solution does not work, since it also requires the training data to be precisely segmented and transcribed for conventional supervised learning-based acoustic models. Segmenting and transcribing manually is extremely time-consuming and costly. The semi-automatic way of performing such preprocessing—applying voice activity detector (VAD) for segmentation and decoding the speech using an already trained acoustic model—may help to some extent, but it lacks reliability and needs additional attention from humans. Therefore, we try to solve the problem by re-recording clean data provided by a trusted speech corpus in various real environments, such as classrooms and telephone channels, as shown in Table 1. In this way, it is possible to acquire paired data to perform various experiments by training acoustic models using different transmission channels’ conditions. We take inspiration for re-recording an existing dataset from the NTIMIT [21] and CTIMIT [22] corpora. They also re-recorded from the original TIMIT acoustic-phonetic speech corpus [23] to utilize the existing transcription for speech recognition task. They are the telephone channel and cellular channel recordings of the TIMIT dataset in the early 1990s. While recording, they also stumble upon unknown noise or artifacts, which are not easy to explain. Therefore, they take some filtering approaches as preprocessing to mitigate the effect. They also pay attention to acquire perfectly aligned utterances with the original recording. In our case, we follow the strategy to acquire data of Japanese language. Even though this method of re-recording is less expensive than recording and transcribing newly recorded data, they still need considerable attention. The lack of synchronization between playing and the recording devices can cause the data to have a misalignment problem. Therefore, the misalignment problem needs to be fixed. There are measures to handle distortions [24] in data. We developed a Euclidean-distance-based alignment correction method. Temporal misalignment of re-recorded data and the method to correct it are described in Section 2.3 and Section 2.3.1.

In this paper, we discuss telephone channel recordings as well as wireless pin mic recordings in a classroom environment. A concatenated long recording of 2 h 19 min consisting of ten monologue speech recordings from the Corpus of Spontaneous Japanese (CSJ) [25], eval1 test dataset is played through a speaker and recorded through different telephone channels as real-world test data. We prepared 26 recordings of about 6 h from the CSJ training set using the mobile LTE channel only for training purposes. The classroom data are recorded using a wireless pin mic by playing 10 monologue speech recordings from CSJ eval3 of about 1 h 40 min. Different playing and recording channels are summarized in Table 1. Landline denotes intercom landline telephone. Mobile 3G and mobile LTE denote recordings using 3rd generation (3G) and 4th generation (4G) LTE cellular networks, respectively. SoftBank carrier is used to record mobile channel data.

#### 2.2.2. Spectral Analysis on Re-Recorded Speech

Figure 1 shows the long-term spectra of re-recorded data through telephone and wireless microphone channels, respectively, as opposed to their original clean counterpart. We notice that the recording data through the mobile LTE channel have higher sensitivity at a lower frequency band than other recording channels. In this research, we focus on improving recognition performance for the LTE channel (called mobile LTE hereinafter) and low-quality wireless pin mic (called pin mic hereinafter), the two most troublesome types of speech to deal with, since they produce the largest character error rate among all the re-recorded categories of data investigated in this research.

### 2.3. Problems Regarding Re-Recorded Speech: Temporal Misalignment

The task of re-recording an audio is performed by playing the audio through a player device and recording the audio simultaneously using a recorder device. In the case of telephone recordings, transmission steps are involved between playing and recording. The start of the recording and playing time often lacks synchronization. Even though it may be possible to start playing and recording at the same time in an ideal situation, there may be different reasons to cause timing mismatch between the pair of events. If the playing device and recording device lack clock synchronization, an incremental delay between estimated start and actual start with time is observed. When using cellular network (mobile phone) channels for transmission in Japan, a delay of up to about 400 ms and also jitter can be experienced depending on the transmission network.

We performed a preliminary analysis on re-recorded wireless pin mic data of a lecture hall. An incremental delay of 20 ms in about every 10 min is noticed in Figure 2. An accumulated delay of 150 ms can be noticed at the start of the 10th lecture. However, the temporal misalignment for telephone recording (CSJ-eval1) is not as simple. Figure 2 shows variable temporal misalignment throughout the re-recording period of 2 h 19 min with the interval of duration of each lecture. We can see a few frames delay at the actual start of the first lecture than the hypothetical start in Figure 2. As this figure shows, time deviations sometimes exceed 200 ms, and similar time deviations were observed in terms of IPU segment units (units separated by silence greater than 200 ms in CSJ [25,26]). Therefore, we propose a correction method in the following sections.

#### 2.3.1. Proposed Misalignment Correction Method Based on Segment-Level Matching with Euclidean Distance

To correct the misalignment, first, a rough starting point $t_{start}^{{}^{\prime }}$ of the re-recorded speech is estimated. Then, a segment-level matching between the pair of segments $x_{t}\ldots x_{t+N-1}$ and $y_{t^{{}^{\prime }}}\ldots y_{t^{{}^{\prime }}+N-1}$ is performed. The optimal starting point $\hat{t}$ is estimated by finding the frame for which the average Euclidean distance is minimum,
(8)$$\hat{t}=\underset{t^{{}^{\prime }}\in \left\{\right|t^{{}^{\prime }}-t\left|\leq D_{max}\right\}}{\arg\min}\left(\frac{1}{N}\sum_{n=0}^{N-1}d\left(x_{t+n},y_{t^{{}^{\prime }}+n}\right)\right).$$
A frame consists of speech data of 10 ms.

Though there are different ways of measuring distortion between speech signals [24], we use the Euclidean distance between MFCC features as distortion measurement. We calculate Euclidean distance between the feature vector of original speech at the $t^{th}$ frame and the feature vector of re-recorded speech at the $t^{{}^{\prime }th}$ frame using Equation (Equation 9).
(9)$$d\left(x_{t},y_{t^{{}^{\prime }}}\right)=\sqrt{\sum_{n}{\left(x_{t,n}-y_{t^{{}^{\prime }},n}\right)}^{2}},$$
where *n* denotes the number of feature dimensions. We assume that $t^{{}^{\prime }}$ falls in the range $t-D_{max},\ldots ,t,\ldots ,t+D_{max}$. $D_{max}$ is the number of frames to search before and after each point of time.

We used a sine wave signal in front of each lecture to indicate starting of individual lectures at the time of concatenating them as a preparation of re-recording to ensure stable quality for the speeches as much as possible. Using the sine wave as guiding point, the starting point of a lecture is first guessed manually. Then, the correct starting point is estimated automatically using the misalignment correction method described above. Then, individual lectures are separated from the long re-recorded speech (segment length to compute Euclidean distance is N=200 frames). In Figure 3, we show the misalignment correction at the starting point in visualized form for re-recorded speech.

#### 2.3.2. Filtering of Re-Recorded Speech

We show in the Figure 4 that there are also internal misalignment in the case of mobile LTE data other than misalignment at the starting points only. This problem was realized after correcting the initial temporal misalignment at the starting point using the Euclidean-distance-based method described in Section 2.3.1. However, since we propose that we train a regression model to learn feature transformation from clean data to real environment data frame by frame, the paired data of the clean-target environment need to be prepared as much as accurately possible. Therefore, we filter out the segments from the re-recorded data, those that do not match the corresponding original speech due to suffering from delay or jitter.

We first add a 200 ms margin in the beginning and the end of each IPU segments [25,26] of clean data and re-recorded data with estimated starting and ending time of IPU, then perform the forced alignment using the DNN-HMM acoustic model. After acquiring forced alignments from both clean and re-recorded data, we remove silences and short pauses from the alignments. This gives us a pair of speech segments to compare length-wise using their phoneme notations and temporal information. In our experiment, first, we derive the segments which have exactly matching lengths for the pair. However, this leaves us with a very small amount of data, about 53 min of data consisting of 291 utterances to be exact, after being filtered out from CSJ training subset (4109 utterances of about 6 h). Therefore, to increase the amount of data, we loosen the strictness of the filtering by allowing 2, 3, 4 and 5 frames (frame = 10 ms) at both ends of the segments to find more matches in turn. In this way, we acquire 694 utterances of 2.1 h in total, which is about 34% of the original recorded data.

### 2.4. Domain Adaptation Using Re-Recorded Speech and DNN-Based Data Augmentation

#### 2.4.1. Task Setting

The neural networks are ideally trained using a large-scale clean database, and they perform very well for clean data of matched condition. However, they do not perform up to expectation when they are applied on real-world data for recognition. Therefore, we propose a method of fine-tuning the baseline model initially trained with a large-scale database with data augmented with the DNN-based augmentation method and otherwise. As the block diagram depicts in Figure 5, we use a small subset from the large-scale database to acquire the re-recorded data. The re-recorded data are then used to train the feature transformer described in Section 2.4.2. Different sets of data are used in the training phase of the feature transformer model and generation phase. We use the transformed features along with clean, re-recorded and simulated data of a fixed set of speakers. We use a variable number of utterances to control the amount of data for fine-tuning the baseline ASR. The fine-tuned ASR model is depicted as the adapted ASR model in Figure 5. It allows better recognition on real-world data.

#### 2.4.2. DNN-Based Data Augmentation Using Feed-Forward Network Architecture

In this research, we train a DNN to perform a nonlinear transformation for features from clean data to simulate real-world recording-like characteristics. This model is denoted as “feature transformer” hereinafter. The feature-transformer takes a *d*-dimensional clean feature vector at frame *t* with a context of *c* frames before and after the central frame $X_{t}=x_{t}-c,\ldots ,x_{t},\ldots ,x_{t}+c$ and outputs $y_{t}$ after performing feature transformation.
(10)$$y_{t}=f_{L}\left(\ldots f_{l}\left(\ldots f_{2}\left(f_{1}\left(X_{t}\right)\right)\right)\right),$$
where $f_{l}$ is the nonlinear transformation function in layer *l* and $y_{t}$ is a *d*-dimensional feature vector. The training of this DNN model is performed by optimizing the mean square error (MSE) objective function to predict feature vectors of corresponding re-recorded speech.

#### 2.4.3. Data Augmentation Approaches

We take various data augmentation approaches to create the most suitable dataset for training the robust baseline, so that it produces the lowest character error rate (CER%) for the test data with real-world acoustic aspects. To create the datasets for training the ASR model, we take general approaches adopted for data augmentation and noise-robust training, as noted in Table 2. First, we apply μ-law encoding to the clean data to simulate landline quality telephone channel distortion. We show that the data with μ-law encoding give better results than not using them for the target domain in Section 3.1.1. Therefore, we use μ-law encoding for all of the baselines. We create another dataset to train Base-Aug3CN-ASR in Table 2, which contains speed- and volume-perturbed clean data with noisy data that contain G.712 filtering. This dataset was created to improve baseline performance of the landline telephone speech. Because of the increased size of clean data, it not only improves performance for landline but also for clean test set. We create another dataset that does not contain any clean data to train the Base-Aug3N-ASR baseline models showed in Table 2. We create this dataset considering the noisy characteristics of the re-recorded test data.

To improve the performance by fine-tuning, we needed to start from an elevated platform. Therefore, keeping real-world scenarios in mind, we perform simulation-based data augmentation. Speed perturbation is used not only to increase the amount of data, but it also gives us two different pitches for each speaker, which somewhat simulates the effect of increasing number of speakers. Therefore, we obtain a three times larger and diverse dataset with the same content, giving us the privilege to use the same transcription for the supervised training of ASR models. Adding volume perturbation allows us to simulate different vocal levels or quality of device. By adding noise, we make the baselines robust to the common noises, which can be experienced in common indoor and outdoor scenarios.

We first experiment on various combinations of data augmentation conditions for the fine-tuning dataset, considering the composition of the baseline datasets. We find the combination of simulation conditions to fine-tune Base-Aug3N-ASR (other baselines did not produced desired result), which produces the best result for most of the cases by carrying out experiments. Therefore, we propose simulation and conditions of datasets for the training adaptation-based experiments. Note that the core clean data for fine-tuning datasets used in our domain adaptation experiments are very small compared with the baseline model. Therefore, using more variations of clean data balances it when re-recorded data and transformed features are used together. We also conducted fine-tuning experiments with speed- and volume-perturbed re-recorded data to see the effect of increased re-recorded data on domain adaptation. In Table 3, since the first six ASR models are intended for fine-tuning for the telephone channel, we use μ-law encoding and filtering with the noisy data. However, FT-P-ASR and FT-PT-ASR are intended for wireless pin mic domain adaptation. Therefore, we do not apply μ-law encoding or any filtering on the fine-tuning datasets.

## 3. Experimental Setup

### 3.1. Datasets

In this section, we gradually describe the datasets and their preparation for training the baselines and fine-tuning them. First, we prepare datasets to train three baseline models for both DNN-HMM ASR and E2E ASR. Then, the data prepared for fine-tuning are described. The proposed fine-tuning dataset contains an element called transformed features (Trans. in Table 3).The transformed features are extracted using the proposed feature transformer model. The dataset for training the feature transformer model is explained in Section 3.1.3. The experiments are performed using subsets of the Corpus of Spontaneous Japanese (CSJ) [25,26] with a sampling rate of 8 kHz.

#### 3.1.1. Datasets for Training Baselines

We prepare datasets with only clean and multi-conditional data for training three baselines for DNN-HMM ASR (Base-NoAug-TDNN, Base-Aug3CN-TDNN and Base-Aug3N-TDNN) and E2E ASR (Base-NoAug-E2E, Base-Aug3CN-E2E and Base-Aug3N-E2E). The dataset used to train Base-NoAug-TDNN and Base-NoAug-E2E consists of 948 academic lectures of CSJ of duration of 233 h, which is the seed (core clean data) for all the baseline training datasets. Training dataset for Base-Aug3CN-ASR contains a total 933 h of data, consisting of three-parts clean data with speed (0.9, 1, 1.1) and volume (factor: 0.7–1.5) perturbation and one-part noisy data created with additive noises chosen from a subset of the noise database “JEIDA-NOISE” [27]. The noise types used are exhibition booth, crowd, computer room (medium), computer room (workstations), air conditioner (large), exhaust fan and air duct. The noises are selected and added randomly with a random SNRs over the range of 5 to 20 dB with a 5 dB interval.The G.712 filter [28] is used to distort the noisy data for telephone channels. Therefore, the dataset used to train Base-Aug3CN-TDNN and Base-Aug3CN-E2E contains four times the clean dataset of duration 933 h. The multi-conditional dataset of 700 h, used to train the Base-Aug3N-TDNN and Base-Aug3N-E2E models, contains three-parts noisy data, prepared by applying speed and volume perturbation, noise and filtering. Therefore, this dataset does not contain any clean data. All of the above datasets are encoded using 8-bit μ-law encoding. We decided to apply μ-law encoding on every dataset by comparing the performances of Base-NoAug-TDNN and Base-NoAug-E2E trained by the dataset with and without μ-law. They are compared in Table 4.

Base-NoAug-ASR trained with TDNN by using clean data of 233 h with μ-law encoding performs better for every kind of test dataset, despite aiming for only mobile variations. We infer that it performs better for clean and landline as well because of the difference in models caused by random initialization. Since we obtained better results for mobile variations by the E2E model trained with data containing μ-law encoding, as expected, we decided to apply μ-law encoding on all of the datasets to perform further experiments.

#### 3.1.2. Dataset for Fine-Tuning

We have 26 re-recorded lectures for LTE domain experiments in total for training purposes using the training subset of CSJ corpus. We use 9 recordings from them to perform fine-tuning for LTE domain. The fine-tuning dataset contains three parts of clean data with speed and volume perturbation, one-part noise and filtering, one-part re-recorded speech and another same combination with one-part transformed features of the same clean content for the matching domain. We change the amount of seed data from 0.2 to 1.5 h for the experiments and compare the results for validating the proposed technique. The term “seed” here denotes the core clean data that are used for data augmentation.

For the wireless pin mic data in the classroom scenario, we only use the 10 clean recordings of the Eval3 test dataset of CSJ and re-record them in the said condition. Since we only acquire 10 recordings of 10 speakers, a total of 1.32 h, we use 9 of the recordings (≈1.2 h on average) for fine-tuning, leaving 1 recording to perform testing. We repeat this process 10 times to do 10-fold cross-validation for all of the the speakers. We do not apply μ-law encoding or G.712 filtering representing telephone channels on simulated data of fine-tuning dataset for the wireless pin mic.

#### 3.1.3. Dataset for Training Feature Transformer Model

We train the feature transformer, as depicted in Figure 6, using the 17 recordings, excluding 9 recordings that are used to generate transformed features for fine-tuning. We use only 553 utterances at most for training the feature transformer model, which we obtain by filtering out temporally mismatched utterances from 17 pairs of clean-re-recorded data. We prepare 5 sets of training pairs with durations of 0.2, 0.5, 1.0 and 1.5 h (equivalent to the size of seed mentioned in Section 3.1.2) from the set by picking a subset of utterances from 17 speakers. We train these models for validating our proposed method. For comparing the performance in general, we use the largest seed of 1.5 h.

In the case of training the feature transformer for wireless pin mic, we use the nine pairs of recordings leaving one for generating transformed feature. We train 10 feature transformer models for 10-fold cross-validation.

### 3.2. Evaluation Tasks

Evaluation is performed on 10 recordings of eval1, which is a clean dataset, called “Clean” along with re-recorded variations of eval1, called “Landline”, “Mobile 3G” and “Mobile LTE” for experiments related to telephone domain adaptation. The evaluation of the wireless pin mic channel’s speech recognition in a classroom environment is performed using variations of eval3 dataset of CSJ, called “Clean” and “wireless pin mic”.

### 3.3. Explanation of Models

#### 3.3.1. DNN-Based Feature Transformer Model

We train the feed-forward type of DNN as the feature transformer which learns nonlinear transformation for the input data to take it closer to the target data. We use DNN with different configuration for LTE and pin mic transformation. We decide the configuration after performing hyperparameter tuning. Both of the models are trained using log-Mel filter bank (F-bank) features of 40 dimensions and pitch features of 3 dimensions, totaling 43 dimesions of input features. We also use first derivative Δ and second derivative ΔΔ of the acoustic features as dynamic features. Per speaker cepstral mean variance normalization (CMVN) is performed to reduce the effect of differences in input. For the LTE feature transformer model, ±5 frames are used as context frames. Therefore, the input layer consists of 1419 nodes. Three hidden layers with 1024 hidden units in each are used. For the pin mic feature transformer model, ±8 frames are used as context frames. The input layer consists of 2193 nodes. Two hidden layers with 1024 hidden units in each are used. Both of the models give us 43 dimensions of transformed features as output.

#### 3.3.2. DNN-HMM ASR Model

We use a TDNN as the acoustic model of the ASR as one of the candidates. The baselines are trained using 43 dimensions of F-bank pitch features. Per speaker CMVN is performed on the input features. This neural network consists of seven hidden layers with the following input context with subsampling: [−5, 5], {−1, 2}, {−3, 3}, {−3, 3}, {−7, 2} and {0}. The output layer consists of 9225 units. A trigram language model is used when decoding. The baselines are then fine-tuned using a smaller amount of simulated and re-recorded data, as well as including feature-transformation-based augmented features with them. To perform experiments regarding the feature transformation model and TDNN-based speech recognition, we use the Kaldi toolkit for speech recognition [29].

#### 3.3.3. End-to-End ASR Model

Baseline hybrid CTC/attention-based end-to-end ASR models are trained using the baseline datasets described previously. 43 dimensions of F-bank and pitch features are used to train the transformer. Global CMVN is applied to the input features also, and the data augmentation method SpecAugment [30] is used on the input features. The encoder part of the model has 12 layers, each consisting of 2048 units. The decoder consists of 6 layers with 2048 units in each of them. A subsampling unit consisting of 2 convolution layers is in the encoder. It reduces the input length to one-fourth. There are four attention heads with 256 dimensions. The wight of α for CTC loss is set to 0.3. The number of output units is 2865, which corresponds to the number of different characters, including Japanese characters. Experiments of end-to-end ASR are performed using ESPnet, the E2E speech processing toolkit [31].

### 3.4. Evaluation Metrices

We use character error rate (CER%) for evaluating the performance of ASR models. CER is denoted by the following equation.
(11)$$\mathrm{CER}=\frac{I+S+D}{N}\times 100=\frac{I+S+D}{C+S+D}\times 100$$
Here, *I* is the number of insertions, *S* is the number of substitutions and *D* is the number of deletions. *C* is the number of correct characters, and *N* is the number of characters in the reference.

We also use character error rate reduction (CERR%), which indicates the improvement when comparing CERs of multiple methods, CER1 and CER2. When CER2 improves from CER1, the CERR (%) of CER2 is calculated using the following equation.
(12)$$\mathrm{CERR}=\frac{CER1-CER2}{CER1}\times 100$$

## 4. Results and Discussion

### 4.1. Results of Domain Adaptation for LTE and Pin Mic Channel When the Largest Amount of Data Are Used

In this section, we compare the baselines and proposed method of fine-tuning with transformed features for the target domain. The starting points we consider are the Base-NoAug-TDNN and BaseA-E2E for DNN-HMM ASR and end-to-end ASR, respectively. We gradually improve the performance by improving the baselines by adding various elements of real environments by simulation. In the following results, the feature transformer model that is used is trained with the largest available data for training (1.5 h of 17 recordings). We mainly propose this method for the DNN-HMM-based system. To also observe the method’s performance, we apply the proposed fine-tuning techniques on the end-to-end model as well. To prove its validity, we perform validation experiments and explain them in the following section with various amounts of data.

In Figure 7, the improvements are shown with the converging character error rate for LTE channel adaptation of TDNN and E2E ASR. We only choose to show the best results for each of the ASR models in the figure. FT-L-TDNN and FT-LT-TDNN in the Figure 7a represent fine-tuning of Base-Aug3N-TDNN with speed- and volume-perturbed clean data along with noisy data accompanying re-recorded speech only, and with adding transformed features to the preceding combination of data, respectively. On the other hand, FT-Aug5L-E2E and FT-Aug5LT-E2E in Figure 7b represent fine-tuning of the Base-Aug3N-E2E model with the same orientation as the TDNN version; the only difference is that in the case of end-to-end ASR, a better result can be observed when larger real-environment data are introduced. Therefore, experiments are performed for threefold and fivefold speed and volume perturbation of re-recorded LTE channel data to find that end-to-end models perform the best when the re-recorded data size is the largest, and the transformed features are the most effective at that time.

In Table 5, we show the performance in detail. The character error rate reduction (27.0%) for the mobile LTE channel is the best for the proposed fine-tuning method with DNN-based data augmentation, prepared with our proposed method of data augmentation for domain adaptation of DNN-HMM ASR. Table 6 shows that end-to-end model-based speech recognition performs better with the adaptation proposed that uses a larger set of augmented re-recorded data. Therefore, character error rate reduction of 36.4% is obtained for FT-Aug5LT-E2E, which is the best improvement found in this research.

Though this research focuses on the purpose of improving recognition performance of real-environment data, we found improvement on the recognition of clean data also when the amount of clean data is used the most. For Base-Aug3CN-ASRs, the CERR is 6.2% for both DNN-HMM ASR and E2E ASR. Though the relative improvement for mobile 3G is better for DNN-HMM ASR, it is the same for both of the methods in the case of landlines. The E2E-based methods start at lower CER to begin with. Though we did not perform training adaptation for landline and mobile 3G, we obtain improvement by considering different real-world conditions, such as distortions and noises, while preparing a better baseline. The improvement strategy reflects on the CER for those channels.

In the case of wireless pin mic recordings in a classroom environment, we observed interesting behavior when performing fine-tuning using the proposed feature transformation-based method of data augmentation. The method is first developed to improve telephone channels and is then applied to the classroom recordings to note its generalization ability. The feature transformation model is trained using fewer data than the feature transformer used for the telephone channel. Moreover, we need to keep in mind that the nature of the test dataset is completely different than that of the eval1 dataset. As expected, it does not work up to expectation. We can see the convergence and divergence in Figure 8. However, we find impressive improvement with CERR of 29.7% for when we use re-recorded data with the simulated data for training adaptation in the case of DNN-HMM ASR in Table 7. For the E2E-based approach, though we fail to achieve the expected result from the proposed method, the best performance is achieved for the Base-Aug3N-E2E, where the data do not contain any clean data.

The results indicate that the adaptation dataset contents fall into the mismatched domain along with the data size issue. Moreover, our investigation shows the variability in the recording quality of data in terms of volume and so on compared with telephone speech. Therefore, consistent feature transformation could not be achieved. Though the CER is the largest to begin with, for a clean test dataset, the best CER is achieved consistently with all the other experiments for Base-Aug3CN-TDNN and Base-Aug3CN-E2E in Table 7 and Table 8, respectively. Because of the restriction in the usability of data, we do not perform validation experiments for classroom wireless pin mic tasks.

### 4.2. Effect of Variability in Recording Quality

In Figure 9, we show the spectral analysis of pin mic re-recordings opposed to their original counterparts. In the figure, we show the spectrum of the recordings divided in two groups according to the sessions they were recorded in. The first session is depicted by red lines and the second session is depicted by blue lines. This analysis shows us the problem of significant difference in level (dB) of re-recorded speech between two recording sessions.

Some re-recordings, for example, the re-recordings of Speaker A, have an amplitude of reasonable audibility. On the other hand, the re-recording set for Speaker B has an amplitude which is hardly audible as opposed to its clean counterpart. In the dataset of 10 speakers, re-recordings of 6 speakers have rather good audibility but are noisy. The rest of the four speakers suffer from audibility problems. Therefore, when we adapt for Speaker A, Speaker B and other poorly audible recordings affect the overall performance, and it continues for 10-fold cross-validation. We suspect that since recording time and settings are different between the group of Speaker A and the group of Speaker B, variability occurred. In the future, we plan to conduct research addressing this problem by applying data augmentation with session-dependent feature transformation models.

In addition to the explanation above, we would like to state that though directly incomparable due to the differences in model configuration, we obtain a better result for eval1 clean (CER 8.4% for Task1 in [16]) by changing CTC weight α from 0.1 to 0.3 comparing to the state-of-the-art method. Though we use smaller training data (233 h comparing with 581 h), the SpecAugment data augmentation technique helps to give it a jump start. Moreover, the data augmentation method adopted for baseline 2 provides variations in different aspects for the same data and helps to produce 5.8% CER for eval1 clean.

### 4.3. Validation Experiments for mobile LTE Channel with Limited Re-Recorded Data

We perform additional experiments to find out the minimum optimal amount of data that need to be prepared for training adaptation, as well as to validate the proposed method, proving its consistency. We acquire the seed amounts of clean data of 0.2, 0.5 and 1.0, and the most is 1.5 h (same condition as Table 5 and Table 6). We perform the detailed experiment only on telephone speech for the DNN-HMM model. The proposed datasets of FT-L-ASR, FT-LT-ASR, FT-Aug3L-ASR, FT-Aug3LT-ASR, FT-Aug5L-ASR and FT-Aug5LT-ASR are compared in Figure 10. Experiments are performed to observe the effect of re-recorded data only on the fine-tuning by increasing the amount of LTE channel re-recorded data in “FT-Aug3L-TDNN” and “FT-Aug3LT-TDNN” by adding speed (0.9, 1 and 1.1) and volume (factor: 0.7–1.5) perturbation to the LTE data. We increase the amount of LTE data even more in “FT-Aug5L-TDNN” and “FT-Aug5LT-TDNN” by adding more speed perturbation (0.8, 0.9, 1, 1.1 and 1.2). Additionally, for each of the combinations, the amount of data used to train the feature transformer model also matches with the size of the seed.

In Figure 10, the effect of re-recorded data itself for fine-tuning is proved. Though adding more re-recorded data helps reduce the distance between fine-tuned TDNN with augmented LTE re-recordings and fine-tuned TDNN with augmented LTE re-recordings along with transformed features (pink–blue, green–purple and orange–red curve pairs in Figure 10a), it does not necessarily improve the whole performance, rather, it represses the models from converging to the smallest character error rate possible. We do not have more data to observe if they are going to decrease drastically or gradually. However, we clearly see the effectiveness of fine-tuning with transformed features in each case. Moreover, in Figure 10b, the models show interesting behavior while the dataset is the smallest and the largest for every model in our task. We notice gradual decrement of the FT-LT-E2E after the point 0.5. We increase the amount of re-recorded data by applying speed and volume perturbation on it, too. In this way, we can observe the effectiveness of the proposed feature transformation method (FT-Aug3LT-E2E and FT-Aug5LT-E2E) with the support of larger simulated re-recorded data. Moreover, with data augmentation for re-recorded speech, the model converges faster even with a smaller seed. The character error rate reduced to 26.1% for the DNN-HMM-based approach (FT-LT-TDNN) and to 34.9% for the end-to-end-based approach (FT-Aug5LT-E2E) by using a seed of 30 min only.

## 5. Conclusions

We conclude this research by summarizing the proposals and the tasks achieved. The main focus of this research was to find a solution for data scarcity in a real-world scenario. We took telephone channel speech and wireless pin mic speech into account. Since it is costly to record data randomly and transcribe them correctly, we propose taking existing data with proper transcription into account and re-record them in the desired environment. The re-recorded data are used to perform feature transformation to create more natural real-environment speech features at a low cost. Our proposed approach of using transformed features with a simple regression model, along with augmented datasets and re-recording itself, improves the overall performance for the mobile LTE channel. Moreover, our detailed investigation shows significant CERR of 26.1% for the DNN-HMM-based approach (FT-LT-TDNN) and 34.9% for the end-to-end-based approach (FT-Aug5LT-E2E) by using a core amount of data of 30 min only. We also show how to handle the misalignment or distortions in re-recorded data by taking simple geometrical approach as a preprocessing technique.

Though we achieve expected performance for the proposed method in the case of mobile LTE channel data, more research is needed to achieve better performance at the same scale in the case of the wireless pin mic channel, since it contains variability in recording quality over different sessions.

Therefore, in the future, we plan to take a session-dependent approach for data with recording quality issues, as well as a self-supervised approach of extracting feature expressions, and change to a feature transformation model to achieve more sophisticated construction of the ANN, such as recurrent neural network (RNN) based sequence-level feature mapping in contrast with the feed-forward neural network approach adopted in this research to prevent temporal dependency and so on for more generalized improvements on various multi-conditional real environments.

## Figures and Tables

**Figure 1 sensors-22-09945-f001:**
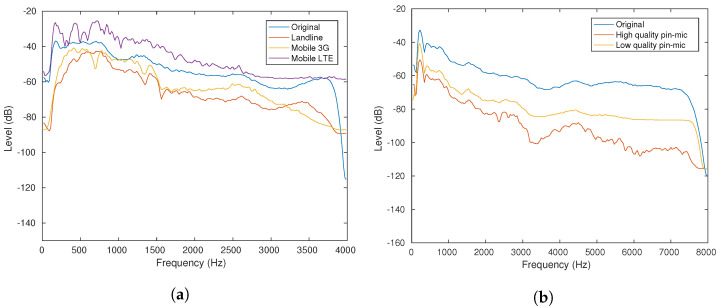
Spectral analysis of original and re-recorded speech. (**a**) Original and re-recorded speech using telephone channels (CSJ eval1 dataset), (**b**) Original and re-recorded speech using wireless pin mic channels (CSJ eval3 dataset).

**Figure 2 sensors-22-09945-f002:**
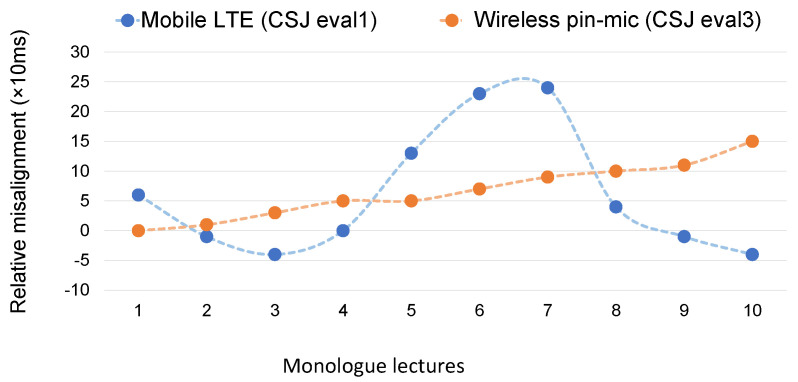
Misalignment analysis of re-recorded speech for mobile LTE and wireless pin mic channels.

**Figure 3 sensors-22-09945-f003:**
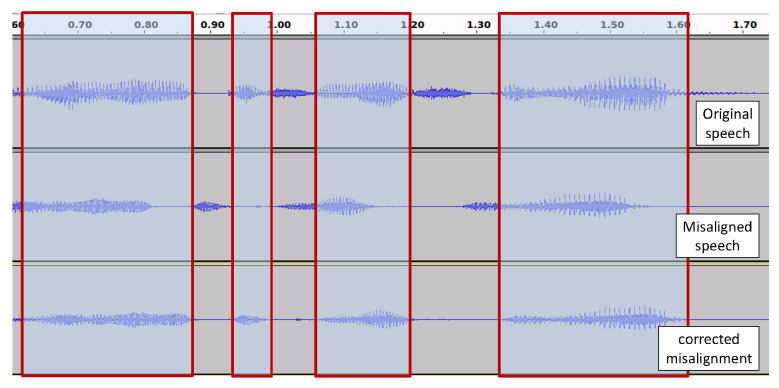
Misalignment correction with proposed Euclidean-distance-based method.

**Figure 4 sensors-22-09945-f004:**
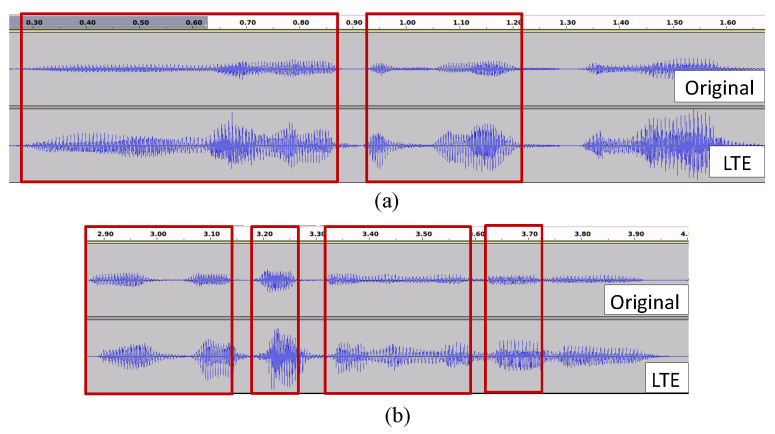
(**a**) Aligned segments of original and mobile LTE re-recorded speech, (**b**) Misaligned segments inside recording those need to be filtered out to train the regression model.

**Figure 5 sensors-22-09945-f005:**
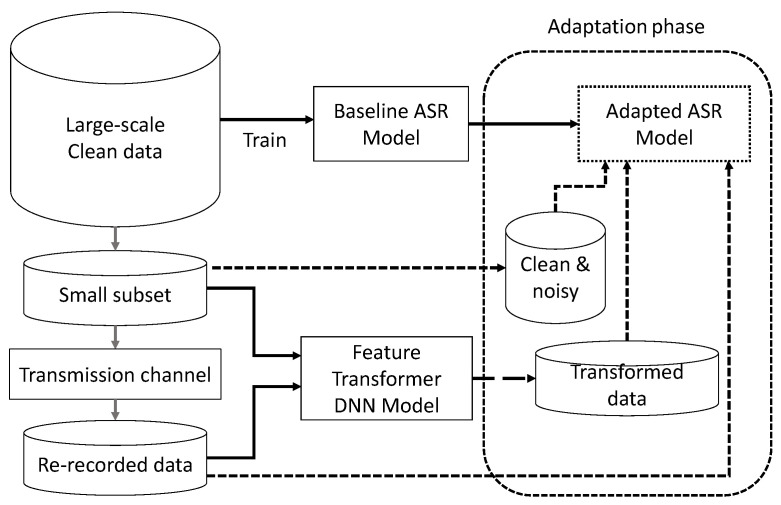
DNN-based data augmentation and adaptation of ASR model.

**Figure 6 sensors-22-09945-f006:**
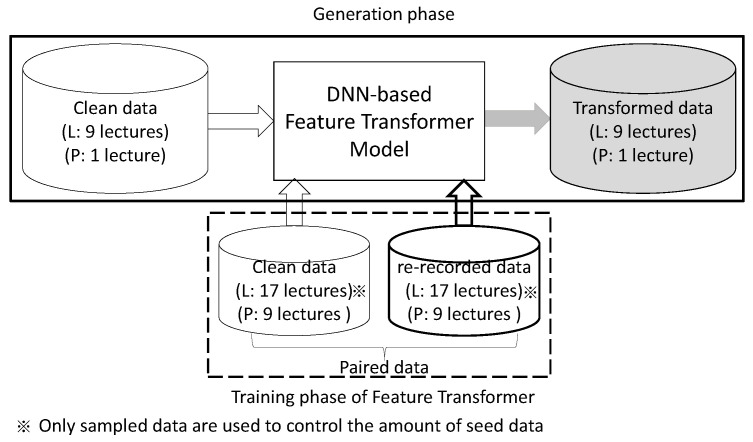
Training of DNN-based feature transformer model.

**Figure 7 sensors-22-09945-f007:**
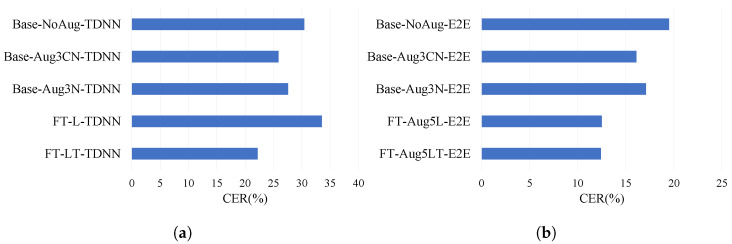
Performance of data augmentation on telephone channel speech. (**a**) TDNN: domain adaptation for LTE, (**b**) E2E: domain adaptation for LTE.

**Figure 8 sensors-22-09945-f008:**
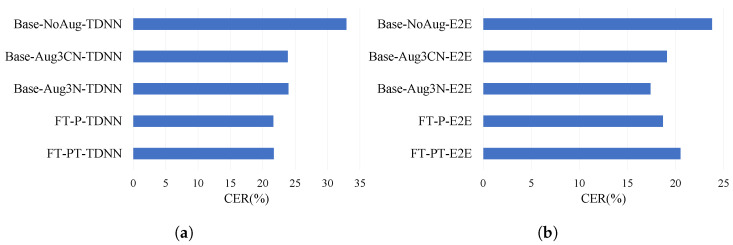
Performance of data augmentation on wireless pin mic speech in a classroom environment. (**a**) TDNN: domain adaptation for pin mic, (**b**) E2E: domain adaptation for pin mic.

**Figure 9 sensors-22-09945-f009:**
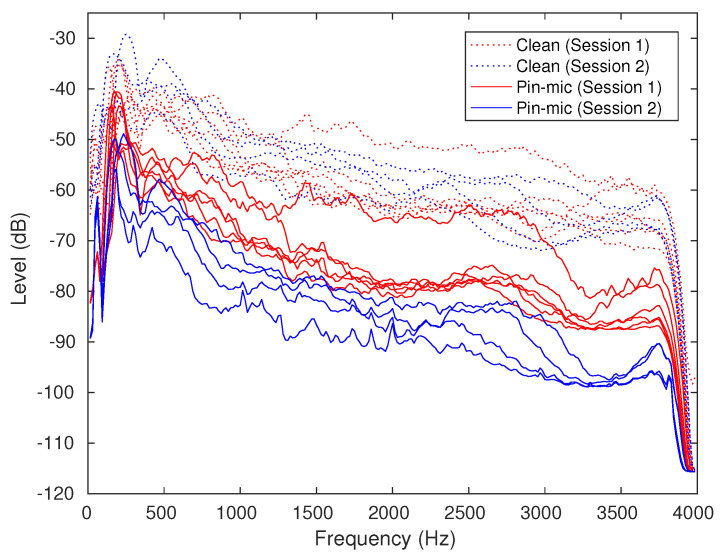
Spectral analysis of wireless original clean data and re-recordings through pin mic channels after down sampling.

**Figure 10 sensors-22-09945-f010:**
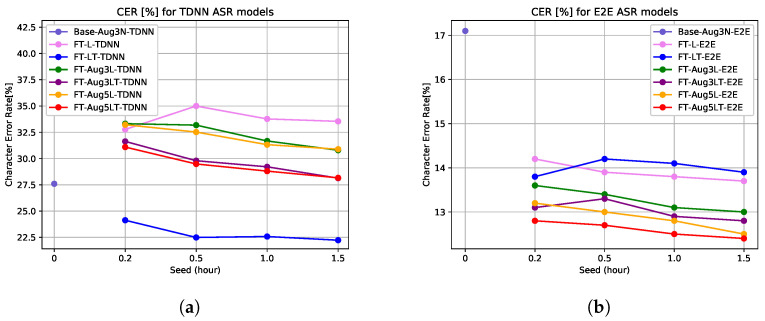
Performance of data augmentation for LTE telephone speech by reducing data size. (**a**) TDNN: domain adaptation for mobile LTE, (**b**) E2E: domain adaptation for mobile LTE.

**Table 1 sensors-22-09945-t001:** Recording devices and transmission channels for re-recorded speech.

Re-Recorded Dataset	Recording Device (mic)	Channel	
		Caller	Receiver
Landline	Landline	Landline	Landline
Mobile 3G	Mobile	SoftBank 3G	SoftBank 3G
Mobile LTE		SoftBank LTE	Landline
Classroom	High-quality pin mic	2.4 GHz digital wireless	
( wireless pin mic)	Low-quality pin mic	800 MHz analog wireless	

**Table 2 sensors-22-09945-t002:** Dataset for training baselines. Notations: S1 = speed perturbation (0.9, 1, 1.1), volume perturbation (0.7–1.5). The notation of “ASR” is replaced in the following section depending on the type of model used (ASR = TDNN or E2E). Size denotes the amount (×233 h) of data. The notations “✓” denotes “applied” and “-” denotes “not applicable”.

Model	Clean				Noisy				Total Size (×233 h)
	μ-Law Encoding	Speed Pert.	Vol Pert.	Size	μ-Law Encoding	Speed Pert.	Vol Pert.	Size	
Base-NoAug-ASR	✓	-	-	1	-	-	-	0	1
Base-Aug3CN-ASR	✓	S1	✓	3	✓	-	-	1	4
Base-Aug3N-ASR	-	-	-	0	✓	✓	✓	3	3

**Table 3 sensors-22-09945-t003:** Dataset for fine-tuning of Base-Aug3N-ASR. Notations: FT= fine-tuning, L = adapted by LTE re-recordings, P = adapted by pin mic re-recordings, S1 = speed perturbation (0.9, 1, 1.1), S2 = speed perturbation (0.8, 0.9, 1, 1.1, 1.2), V = volume perturbation (0.7–1.5), F = G.712 filter, T = transformed features. The notation of “ASR” is replaced in the following section depending on the type of model used (ASR=TDNN or E2E). Size denotes the number of seed-sized datasets used for fine-tuning. “Seed” denotes the amount of core clean data. Seed (L) = {0.2, 0.5, 1, 1.5} h; seed (P) = ≈1.2 h. The notations “✓” denotes “applied” and “-” denotes “not applicable”.

Model	Clean (μ-Law, S1, V)	Noisy (μ-Law, F)	Re-Recorded (μ-Law)			Trans.		Total Size (×Seed)
	Size	Size	SP.	Vol.	Size	T	Size	
FT-L-ASR	3	1	-	-	1	-	0	5
FT-LT-ASR	3	1	-	-	1	✓	1	6
FT-Aug3L-ASR	3	1	S1	✓	3	-	0	7
FT-Aug3LT-ASR	3	1	S1	✓	3	✓	1	8
FT-Aug5L-ASR	3	1	S2	✓	5	-	0	9
FT-Aug5LT-ASR	3	1	S2	✓	5	✓	1	10
FT-P-ASR (no μ-law & F)	3	1	-	-	1	-	0	5
FT-PT-ASR (no μ-law & F)	3	1	-	-	1	✓	1	6

**Table 4 sensors-22-09945-t004:** Telephone speech: character error rate (CER%) of TDNN and E2E ASR models trained by data with or without μ-law encoding.The notation “✓” denotes “applied” and “×” denotes “did not apply”. The Expressions in bold font denotes the baselines models those are trained with μ-law encoding. The bold numbers represent the smallest word error rate comparing results between baselines with or without μ-law encoding for each category of ASR (TDNN or E2E).

Model	μ-Law Encoding	Test Dataset (CSJ eval1)			
		Clean	Re-Recorded		
			Landline	Mobile 3G	Mobile LTE
Base-NoAug-TDNN	×	9.5	11.1	24.4	31.5
**Base-NoAug-TDNN**	✓	**9.4**	**11.0**	**23.6**	**30.6**
Base-NoAug-E2E	×	**6.2**	**6.8**	15.3	20.6
**Base-NoAug-E2E**	✓	6.3	7.0	**14.4**	**19.5**

**Table 5 sensors-22-09945-t005:** Telephone speech: character error rate (CER%) of different TDNN ASR models and character error reduction (CERR%) from Base-NoAug-TDNN. The notation “-” denotes “not applicable”. The expressions with bold font represent the models trained with proposed fine-tuning method. The bold numbers represent the best result for each test dataset.

Model	Data Size for Training/ Adaptation (Seed) (h)	Test Dataset (CSJ eval1)							
		Clean		Re-Recorded					
				Landline		Mobile 3G		Mobile LTE	
		**CER**	**CERR**	**CER**	**CERR**	**CER**	**CERR**	**CER**	**CERR**
Base-NoAug-TDNN	233 (233)	9.4	-	11.0	-	23.6	-	30.4	-
Base-Aug3CN-TDNN	933 (233)	**8.8**	**6.2**	**9.7**	**11.5**	18.5	21.6	25.9	27.8
Base-Aug3N-TDNN	700 (233)	9.6	−1.8	9.9	10.2	**17.1**	**27.8**	27.6	9.4
FT-L-TDNN	7.5 (1.5)	-	-	-	-	-	-	33.6	−10.2
**FT-LT-TDNN**	9 (1.5)	-	-	-	-	-	-	**22.2**	**27.0**
FT-Aug3L-TDNN	10.5 (1.5)	-	-	-	-	-	-	30.8	−1.1
**FT-Aug3LT-TDNN**	12 (1.5)	-	-	-	-	-	-	28.1	7.6
FT-Aug5L-TDNN	13.5 (1.5)	-	-	-	-	-	-	30.9	−1.5
**FT-Aug5LT-TDNN**	15 (1.5)	-	-	-	-	-	-	28.2	7.5

**Table 6 sensors-22-09945-t006:** Telephone speech: character error rate (CER%) of different E2E ASR models and character error reduction (CERR%) from Base-NoAug-E2E. The notation “-” denotes “not applicable”. The expressions with bold font represent the models trained with proposed fine-tuning method. The bold numbers represent the best result for each test dataset.

Model	Data Size for Training/ Adaptation (Seed) (h)	Test Dataset (CSJ eval1)							
		Clean		Re-Recorded					
				Landline		Mobile 3G		Mobile LTE	
		**CER**	**CERR**	**CER**	**CERR**	**CER**	**CERR**	**CER**	**CERR**
Base-NoAug-E2E	233 (233)	6.3	-	7.0	-	14.4	-	19.5	-
Base-Aug3CN-E2E	933 (233)	**5.8**	**6.2**	**6.2**	**11.5**	11.7	18.8	16.1	15.1
Base-Aug3N-E2E	700 (233)	6.5	−3.2	6.7	−4.5	**11.5**	**20.1**	17.1	12.3
FT-L-E2E	7.5 (1.5)	-	-	-	-	-	-	13.7	29.7
**FT-LT-E2E**	9 (1.5)	-	-	-	-	-	-	13.9	28.7
FT-Aug3L-E2E	10.5 (1.5)	-	-	-	-	-	-	13.0	33.3
**FT-Aug3LT-E2E**	12 (1.5)	-	-	-	-	-	-	12.8	34.4
FT-Aug5L-E2E	13.5 (1.5)	-	-	-	-	-	-	12.5	35.9
**FT-Aug5LT-E2E**	15 (1.5)	-	-	-	-	-	-	**12.4**	**36.4**

**Table 7 sensors-22-09945-t007:** Wireless pin mic speech: character error rate (CER%) of different TDNN ASR models and character error reduction (CERR%) from Base-NoAug-TDNN. The notation “-” denotes “not applicable”. The expressions with bold font represent the models trained with proposed fine-tuning method. The bold numbers represent the best result for each test dataset.

Model	Data Size for Training/ Adaptation (Seed) (h)	Test Dataset (CSJ eval3)			
		Clean		Re-Recorded	
				Wireless pin mic	
		**CER**	**CERR**	**CER**	**CERR**
Base-NoAug-TDNN	233 (233)	10.6	-	30.8	-
Base-Aug3CN-TDNN	933 (233)	**10.1**	**6.2**	22.1	2.8
Base-Aug3N-TDNN	700 (233)	11.2	-9.4	22.5	26.7
FT-P-TDNN	6 (1.2)	-	-	**21.6**	**29.7**
**FT-PT-TDNN**	≈7 (1.2)	-	-	21.7	29.4

**Table 8 sensors-22-09945-t008:** Wireless pin mic speech: character error rate (CER%) of different E2E ASR models and character error reduction (CERR%) from Base-NoAug-E2E. The notation “-” denotes “not applicable”. The expressions with bold font represent the models trained with proposed fine-tuning method. The bold numbers represent the best result for each test dataset.

Model	Data Size for Training/ Adaptation (Seed) (h)	Test Dataset (CSJ eval3)			
		Clean		Re-Recorded	
				Wireless pin mic	
		**CER**	**CERR**	**CER**	**CERR**
Base-NoAug-E2E	233 (233)	10.8	-	23.8	-
Base-Aug3CN-E2E	933 (233)	**10.0**	**7.4**	19.1	19.7
Base-Aug3N-E2E	700 (233)	11.4	-5.3	**17.4**	**26.9**
FT-P-E2E	6 (1.2)	-	-	18.7	21.4
**FT-PT-E2E**	≈7 (1.2)	-	-	20.5	13.9

## Data Availability

Not applicable.

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
