# Peer review of "Domain Adaptation with Augmented Data by Deep Neural Network Based Method Using Re-Recorded Speech for Automatic Speech Recognition in Real Environment"

_sensors, 2022, doi:10.3390/s22249945_

Round 1
Reviewer 1 Report
Manuscript ID: sensors-2072564
This manuscript shows a comparative investigation of different channel adaptation methods for real-world speech recognition. The manuscript has reliable data and is well structured. The several comments are shown as follows:
(1)In the Abstract, please explain the word “LTE channel”.
(2)In the Introduction, the extensive use of “We” does not emphasize the content of the study. Passive expressions are recommended.
(3)Please explain the notes “√”, “×”, and “-” in figure 3 and figure 4.
Author Response
Thank you very much for your kind review. Please find the responses in the attachment.

Reviewer 2 Report
TITLE
The title is too long and the interest of the reader is lost. It can be reduced and be specific in the main contribution of the manuscript.
Abstract
The abstract of this manuscript is clear and concrete on the developed topic. The information contained in this section is correctly structured and supported by the information developed in the research that supports this manuscript. It is recommended to improve the explanation about the use of an augmented database to improve the training of the ASR model. Since the explanation given about the cost of acquiring real data to understand the model is generic and technical arguments are not given.
Introduction
The introduction of this manuscript presents the general aspects on which the research is based. but it is necessary to improve the quality of the writing, since some sentences are not correctly written.
In line 56, the quote that is made about reference 10 is not correctly written. The wording of this sentence should be improved.
Rewrite the sentence that begins on line 57, since it presents errors when presenting the reference on which said sentence is based.
It is important to review the writing of the text in the introduction section, some sentences can be observed that cannot be understood properly since references are cited inappropriately grammatically.
At the end of this section it is described how the manuscript is integrated, but this is not according to the journal's guidelines. Which causes that the manuscript cannot be evaluated according to the canons of the journal.
Materials and Methods
This section is not defined as marked by the Journal Guidelines, so it is suggested that they adhere to the format.
Results
This section is not defined as marked by the Journal Guidelines, so it is suggested that they adhere to the format.
The results presented in this manuscript are interesting to describe the performance of a deep neural network (DNN) over an artificial neural network (ANN) using an augmented database. But it is important to follow the journal's guidelines so that it can be correctly evaluated by the reviewers.
Discussion
This section is not individually defined in the writing as marked by the Journal's guidlines. Authors are advised to adhere to the Journal format.
Conclusion
The information presented in the conclusions of the manuscript is clear and concise. It adequately describes the experimental information developed in the research described in the manuscript. The conclusions have a solid basis regarding the premise of the manuscript and the experimental information presented.
References
The distribution of references is adequate. The distribution of references is 34.37% older than 5 years, 34.37% between 5-10 years and the remaining 31.26% older than 10 years.
Author Response

(The authors gave the same response as above.)

Reviewer 3 Report
Very good paper. I just would like to ask authors to add more details related to their mathematical presentation. Specifically, the background of equations 1-3. Finally, authors should describe their future work directions.
Author Response

(The authors gave the same response as above.)

Reviewer 4 Report
The work presented in this manuscript is very interesting but there are several mistakes which are highlighted in the PDF file. The authors are requested to check each comment and correct their manuscript accordingly. The PDF file in which comments are added is attached here

Author Response

(The authors gave the same response as above.)
